# Quantifying and Learning Linear Symmetry-Based Disentanglement

## Abstract

The definition of Linear Symmetry-Based Disentanglement (LSBD) formalizes the notion of linearly disentangled representations, but there is currently no metric to quantify LSBD. Such a metric is crucial to evaluate LSBD methods and to compare to previous understandings of disentanglement. We propose $\mathcal{D}_{\mathrm{LSBD}}$, a mathematically sound metric to quantify LSBD, and provide a practical implementation. Furthermore, from this metric we derive LSBD-VAE, a semi-supervised method to learn LSBD representations. We demonstrate the utility of our metric by showing that (1) common VAE-based disentanglement methods don't learn LSBD representations, (2) LSBD-VAE as well as other recent methods *can* learn LSBD representations, needing only limited supervision on transformations, and (3) various desirable properties expressed by existing disentanglement metrics are also achieved by LSBD representations.

## 1 Introduction

Learning low-dimensional representations that disentangle the underlying factors of variation in data is considered an important step towards interpretable machine learning with good generalization. To address the fact that there is no consensus on what disentanglement entails and how to formalize it, Higgins et al. (2018) propose a formal definition for Linear Symmetry-Based Disentanglement, or LSBD, arguing that underlying real-world symmetries give exploitable structure to data.

However, there is currently no metric to quantify LSBD. Such a metric is crucial to properly evaluate methods aiming to learn LSBD representations and to relate LSBD to previous definitions of disentanglement. Although previous works have evaluated LSBD by measuring performance on downstream tasks (Caselles-Dupré et al., 2019) or by measuring specific traits related to LSBD (Painter et al., 2020; Quessard et al., 2020), none of these evaluation methods directly quantify LSBD according to its well-formalized definition.

We propose $\mathcal{D}_{\mathrm{LSBD}}$, a well-formalized and generally applicable metric that quantifies the level of LSBD in learned data representations. We show an intuitive justification of this metric, as well as its theoretical derivation. We also provide a practical implementation to compute $\mathcal{D}_{\mathrm{LSBD}}$ for common symmetry groups. Furthermore, we show that our metric formulation can be used to derive a semi-supervised method to learn LSBD representations, which we call LSBD-VAE. To make LSBD-VAE more widely applicable, we also demonstrate how to disentangle symmetric properties from other non-symmetric properties, and how to quantify this disentanglement with $\mathcal{D}_{\mathrm{LSBD}}$.

We show the utility of $\mathcal{D}_{\mathrm{LSBD}}$ by quantifying LSBD in a number of settings, for a variety of datasets with underlying $\mathrm{SO}(2)$ symmetries and other non-symmetric properties. First, we evaluate common VAE-based disentanglement methods and show that most don't learn LSBD representations. Second, we evaluate LSBD-VAE and other recent methods that specifically target LSBD, showing that they

36 *can* obtain much better $\mathcal{D}_{\mathrm{LSBD}}$ scores while needing only limited supervision on transformations.
37 Third, we compare $\mathcal{D}_{\mathrm{LSBD}}$ with existing disentanglement metrics, showing that various desirable
38 properties expressed with these metrics are also achieved by LSBD representations.

## 2   Related Work

40 Plenty of works have focused on learning and quantifying disentangled representations recently, but
41 research has shown that there is little consensus about the exact definition of disentanglement and
42 methods often do not achieve it as well as they proclaim (Locatello et al., 2019). To introduce some
43 much-needed formalization, Higgins et al. (2018) proposed to define disentanglement with respect
44 to symmetry transformations acting on the data. They used group theory to provide two formal
45 definitions, which we refer to as (Linear) Symmetry-Based Disentanglement, or (L)SBD. In this
46 paper we focus only on LSBD, not SBD.

47 Several methods have been proposed to learn LSBD representations (Caselles-Dupré et al., 2019;
48 Painter et al., 2020; Quessard et al., 2020). These methods also learn to represent the transformations
49 acting on the input data, assuming various levels of supervision on these transformations. Other
50 methods have previously focused on capturing transformations of the data outside the context of
51 disentanglement as well (Cohen and Welling, 2015; Sosnovik et al., 2019; Worrall et al., 2017).

## 3   Linear Symmetry-Based Disentanglement

53 Higgins et al. (2018) provide a formal definition of linear disentanglement that connects symmetry
54 transformations affecting the real world (from which data is observed) to the internal representations
55 of a model. The definition is grounded in concepts from *group theory*, we provide a more detailed
56 description of these concepts in the Supplementary Material.

57 The definition[1] considers a group $G$ of symmetry transformations acting on the *data space* $X$ through
58 the group action $\cdot : G \times X \to X$. In particular, $G$ can be decomposed as the direct product of $K$
59 groups $G = G_1 \times \ldots \times G_K$. A model's internal representation of data is modeled with the *encoding*
60 function $h : X \to Z$ that maps data to the *embedding space* $Z$. The definition for Linearly Symmetry-
61 Based Disentangled (LSBD) representations then formalizes the requirement that a model's encoding
62 $h$ should reflect and disentangle the transformation properties of the data, and that the transformation
63 properties of the model's encoding should be linear. The exact definition is as follows:

**Definition: Linear Symmetry-Based Disentanglement (LSBD)**   A model's encoding map $h :$
65 $X \to Z$, where $Z$ is a vector space, is LSBD with respect to the group decomposition $G =$
66 $G_1 \times \ldots \times G_K$ if

67   1. there is a decomposition of the embedding space $Z = Z_1 \oplus \ldots \oplus Z_K$ into $K$ vector
68      subspaces,

69   2. there are group representations for each subgroup in the corresponding vector subspace
70      $\rho_k : G_k \to \mathrm{GL}(Z_k), k \in \{1, \ldots, K\}$

71   3. the group representation $\rho : G \to \mathrm{GL}(Z)$ acts on $Z$ as

$$\rho(g) \cdot z = (\rho_1(g_1) \cdot z_1, \ldots, \rho_K(g_K) \cdot z_K), \tag{1}$$

72      for $g = (g_1, \ldots, g_K) \in G$ and $z = (z_1, \ldots, z_K) \in Z$ with $g_k \in G_k$ and $z_k \in Z_k$.

73   4. the map $h$ is *equivariant* with respect to the actions of $G$ on $X$ and $Z$, i.e. , for all $x \in X$
74      and $g \in G$ it holds that $h(g \cdot x) = \rho(g) \cdot h(x)$.

75 Furthermore, we say that a group representation $\rho$ is *linearly disentangled* with respect to the group
76 decomposition $G = G_1 \times \ldots \times G_K$ if it satisfies criteria 1 to 3 from the LSBD definition above.

---

[1]The original definition actually considers an additional set of world states $W$, but our definition is more
practical and can be shown to be the same under mild conditions, see Supplementary Material.

## 4 Quantifying LSBD: $\mathcal{D}_{\mathrm{LSBD}}$

### 4.1 Intuition: Measuring Equivariance with Dispersion

To motivate our metric, let's first assume a setting in which a suitable *linearly disentangled* group representation $\rho$ is known. Let's further assume that the dataset of observations can be expressed with respect to $G$ acting on some base point $x_0 \in X$, i.e. $\{x_n\}_{n=1}^N = \{g_n \cdot x_0\}_{n=1}^N$. Formally, this assumes that the action of $G$ on $X$ is *regular*. In this case, we can use the inverse group elements $g_n^{-1}$ to transform each data point toward the base point $x_0$, i.e.

$$x_0 = g_1^{-1} \cdot x_1 = \ldots = g_N^{-1} \cdot x_N. \quad (2)$$

Since $\rho$ is *linearly disentangled*, we only need to measure the *equivariance* of the encoding map $h$ to quantify LSBD. Equivariance is achieved when $h(g \cdot x) = \rho(g) \cdot h(x)$, for all $g \in G, x \in X$. Given the dataset described above, we can check this property for $x \in \{x_n\}_{n=1}^N$ and $g \in \{g_n\}_{n=1}^N$.[2] In particular, from Equation (2) we can see that we have equivariance if

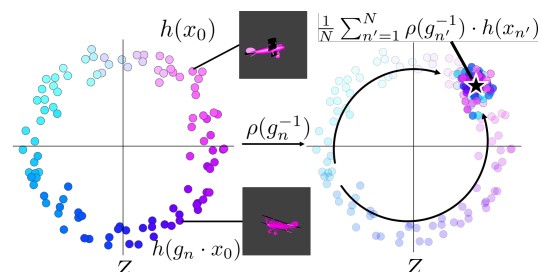

Figure 1: A dataset of images from a rotating object expressed in terms of the group $G = \mathrm{SO}(2)$ acting on a base image $x_0$. It is possible to quantify the level of LSBD of an encoding map $h$ by measuring its equivariance with respect to a group representation $\rho$. Since all data has been generated from $x_0$, equivariance can be measured as the dispersion of the points $\{\rho(g_n^{-1}) \cdot h(x_n)\}_{n=1}^N$.

$$h(x_0) = \rho(g_1^{-1}) \cdot h(x_1) = \ldots = \rho(g_N^{-1}) \cdot h(x_N). \quad (3)$$

This not only characterizes perfect equivariance, but also allows for an efficient way to quantify how close we are to true equivariance, by measuring the *dispersion* of the points $\{\rho(g_n^{-1}) \cdot h(x_n)\}_{n=1}^N$.[3] Given a suitable norm $\|\cdot\|_Z$ in $Z$, we can thus quantify LSBD in this setting as

$$\frac{1}{N} \sum_{n=1}^N \left\| \rho(g_n^{-1}) \cdot h(x_n) - \frac{1}{N} \sum_{n'=1}^N \rho(g_{n'}^{-1}) \cdot h(x_{n'}) \right\|_Z^2, \quad (4)$$

i.e. we compute the mean of $\{\rho(g_n^{-1}) \cdot h(x_n)\}_{n=1}^N$ and use the average squared distance to this mean for points in $\{\rho(g_n^{-1}) \cdot h(x_n)\}_{n=1}^N$ as our LSBD metric, see Figure 1.

However, this formulation requires knowing the right *linearly disentangled* group representation and a suitable norm in $Z$. Moreover, it implicitly assumes a uniform probability measure over the group elements $\{g_n\}_{n=1}^N$. In the next section we formulate our metric for a more general setting.

### 4.2 $\mathcal{D}_{\mathrm{LSBD}}$: A Metric for LSBD

Generalizing the ideas from the previous section with concepts from *measure theory*, we propose a metric to measure the level of LSBD of any encoding $h : X \to Z$ given a data probability measure $\mu$ on $X$, provided that $\mu$ can be written as the pushforward $G_X(\cdot, x_0)_{\#}\nu$ of some probability measure $\nu$ on $G$ by the function $G_X(\cdot, x_0)$ for some base point $x_0$. More formally,

$$\mu(A) = G_X(\cdot, x_0)_{\#}\nu(A) = \nu(\{g \in G \mid G_X(g, x_0) \in A\}), \quad (5)$$

for Borel subsets $A \subset X$. Note that this is only possible if the action $G_X$ is *transitive*.

For example, the situation of a dataset with $N$ datapoints $\{x_n\}_{n=1}^N = \{g_n \cdot x_0\}_{n=1}^N$ corresponds to the case in which $\nu$ and $\mu$ are empirical measures on the group $G$ and data space $X$, respectively:

$$\nu := \frac{1}{N} \sum_{i=1}^N \delta_{g_i}, \qquad \mu := \frac{1}{N} \sum_{i=1}^N \delta_{x_i}. \quad (6)$$

---

[2] Note that $\{g_n\}_{n=1}^N$ can be used to describe all known group transformations between elements in the dataset by means of composition and inverses, since $x_i = g_i \cdot (g_j^{-1} \cdot x_j)$. Thus it suffices to check equivariance for these $N$ group transformations.

[3] Note that we do not actually need to know $x_0$ nor $h(x_0)$.

We define the metric $\mathcal{D}_{\text{LSBD}}$ for an encoding $h$ and a measure $\mu$ as

$$\mathcal{D}_{\text{LSBD}} := \inf_{\rho \in \mathcal{P}(G,Z)} \int_G \left\| \rho(g)^{-1} \cdot h(g \cdot x_0) - M_{\rho,h,x_0} \right\|_{\rho,h,\mu}^2 d\nu(g),$$

$$\text{with } M_{\rho,h,x_0} = \int_G \rho(g')^{-1} \cdot h(g' \cdot x_0) d\nu(g'),$$

(7)

where the norm $\|\cdot\|_{\rho,h,\mu}$ is a Hilbert-space norm depending on the representation $\rho$, the encoding map $h : X \to Z$, and the data measure $\mu$. More details of this norm can be found in the Supplementary Material. Moreover, $\mathcal{P}(G,Z)$ denotes the set of *linearly disentangled representations* of $G$ in $Z$. Lower values of $\mathcal{D}_{\text{LSBD}}$ indicate better disentanglement, zero being optimal.

## 4.3 Practical Computation of $\mathcal{D}_{\text{LSBD}}$

There are two main challenges for computing the metric of Equation (7). First, to calculate the integrals in the formula, all possible datapoints that can be expressed as $g \cdot x_0$ with $g \in G = G_1 \times \cdots \times G_K$ must be available. Second, the infimum of the integrals over all possible linearly disentangled representations must be estimated. This requires finding the possible invariant subspaces $Z = Z_1 \oplus \cdots \oplus Z_K$ induced by the encoding $h$ over which the group representations are disentangled.

We present a practical implementation of an upper bound to $\mathcal{D}_{\text{LSBD}}$ for an encoding function $h$ given a dataset $\mathcal{X}$ generated by some known group transformations. In particular, this approximation of $\mathcal{D}_{\text{LSBD}}$ is designed for a group decomposition $G = G_1 \times \cdots \times G_K$ where each $G_k = \text{SO}(D_k)$ with $k \in \{1, \ldots, K\}$ the group of rotations in $D_k$ dimensions. This implementation approximates the integrals of Equation (7) by using the empirical distribution of $\mathcal{X}$. The invariant subspaces of $Z$ to the subgroup actions are found by applying a suitable change of basis. In the new basis, the disentangled group representations are expressed in a parametric form whose parameters are optimized to find the tightest bound to $\mathcal{D}_{\text{LSBD}}$. See Figure 2 for an intuitive description of the process.

Assume there is a dataset $\mathcal{X}$ that can be modeled in terms of the group decomposition $G = G_1 \times \cdots \times G_k$. For each $G_k$ subgroup there is a set of known group elements $\mathcal{G}_k \subseteq G_k$ uniformly sampled such that the dataset is described in terms of all elements in $\mathcal{G} = \mathcal{G}_1 \times \cdots \times \mathcal{G}_K$ and a base point $x_0$ as $\mathcal{X} = \{(g_1, \ldots, g_K) \cdot x_0 | g_k \in \mathcal{G}_k, \ k \in \{1, \ldots, K\}\}$.

For each subgroup $G_k$ we construct a set of encoded data $\mathcal{Z}_k \subseteq Z$ whose variability should only depend on the action of $G_k$. The set $\mathcal{Z}_k$ is given by $\mathcal{Z}_k = \{z_k(g_1, \ldots, g_K) | g_j \in \mathcal{G}_j, j \in \{1, \ldots, K\}\}$, in which

$$z_k(g_1, \ldots, g_K) = h((g_1, \ldots, g_K) \cdot x_0) - \frac{1}{|\mathcal{G}_k|} \sum_{g' \in \mathcal{G}_k} h((g_1, \ldots, g_{k-1}, g', g_{k+1}, \ldots, g_K) \cdot x_0). \quad (8)$$

Similar to (Cohen and Welling, 2014), we find a suitable change of basis that exposes the invariant subspace $Z_k$ corresponding to the $k$-th subgroup $G_k$. The new basis is obtained from the eigenvectors resulting from applying Principal Component Analysis (PCA) to $\mathcal{Z}_k$. Each element in $\mathcal{Z}_k$ is projected into the first $D_k$ eigenvectors. The new set is denoted as $\mathcal{Z}'_k \subseteq \mathbb{R}^{D_k}$ with elements $z'_k(g_1, \ldots, g_K) \subseteq \mathbb{R}^{D_k}$ that are the projected versions of $z_k(g_1, \ldots, g_K)$.

(Quessard et al., 2020) describes how one could parameterize the subgroup representations of $SO(D_k)$ for arbitrary $D_k$ but here we will focus on $G_k = SO(2)$. In this case, we can parameterize each subgroup representation in terms of a single integer parameter $\omega \in \mathbb{Z}$ as $\rho_{k,\omega}(g_k)$ corresponding to a $2 \times 2$ rotation matrix whose angle of rotation is $\omega$ multiplied by the known angle associated to the group element $g_k \in G_k = \text{SO}(2)$. For this subgroup we can approximate the $M_{\rho,h,x_0}$ from Equation (7) as $M_{k,\omega}$ given by

$$M_{k,\omega} = \frac{1}{|\mathcal{G}|} \sum_{(g_1, \ldots, g_K) \in \mathcal{G}} \rho_{k,\omega}(g_k^{-1}) \cdot z'(g_1, \ldots, g_K). \quad (9)$$

Similar to Equation (7) we would like to find the optimal $\rho_{k,\omega}$ that minimizes the integral over the group representations. We can define a parameter search space $\Omega \subseteq \mathbb{Z}$, e.g. $\Omega = [-10, 10]$ for finding the optimal $\omega \in \Omega$ that minimizes the dispersion, this is expressed in the following equation

$$\mathcal{D}_{\text{LSBD}}^{(k)} = \min_{\omega \in \Omega} \frac{1}{|\mathcal{G}|} \sum_{(g_1, \ldots, g_K) \in \mathcal{G}} \| \rho_{k,\omega}(g_k^{-1}) \cdot z'(g_1, \ldots, g_K) - M_{k,\omega} \|^2. \quad (10)$$

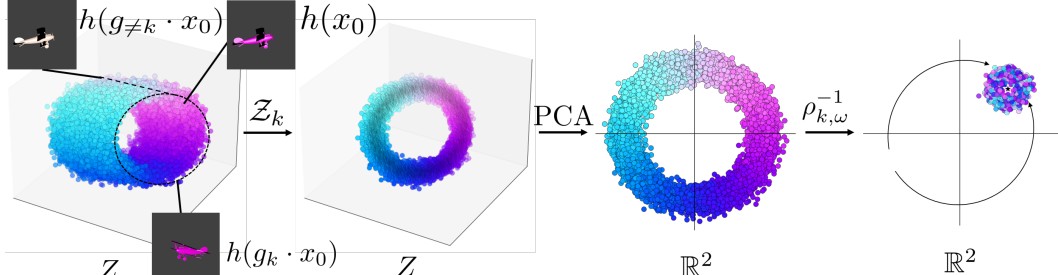

Figure 2: Consider a dataset modeled by a group decomposition $G = G_1 \times \cdots \times G_K$ acting on $x_0$ and is embedded in a latent space $Z$ via $h$. In this example the subgroup $G_k = \mathrm{SO}(2)$ models the rotations of an airplane. Other subgroups $G_{\neq k}$ could also be acting e.g. changes in airplane color. The first step to calculate the disentanglement of $G_k$ is to construct a set of data embeddings $\mathcal{Z}_k \subseteq Z$ whose variability is due to $G_k$. These embeddings are then projected into a 2-dimensional space through PCA. For these projected embeddings we can describe the group representations in a simple parametric form $\rho_{k,w}$. For a given $\rho_{k,w}$ the equivariance of $G_k$ is measured as the dispersion after applying the action of the inverse group representation $\rho_{k,w}^{-1}$.

Each $\mathcal{D}_{\mathrm{LSBD}}^{(k)}$ measures the degree of equivariance of the projected embeddings for each $k$-th subgroup corresponding to the best fitting group representation. The upper bound to the metric is finally obtained by averaging across all subgroups $\mathcal{D}_{\mathrm{LSBD}} \leq \frac{1}{K} \sum_{k=1}^{K} \mathcal{D}_{\mathrm{LSBD}}^{(k)}$.

## 5  Learning LSBD Representations: LSBD-VAE

In this section we present LSBD-VAE, a semi-supervised VAE-based method to learn LSBD representations. The main idea is to train an unsupervised Variational Autoencoder (VAE) (Kingma and Welling, 2014; Rezende et al., 2014) with a suitable latent space topology, and use our metric as an additional loss term for batches of transformation-labeled data.

**Assumptions**  LSBD-VAE requires some knowledge about the group structure $G$ that is to be disentangled. Concretely, the group and its decomposition $G = G_1 \times \ldots \times G_K$ should be known, as well as a suitable *linearly disentangled* group representation $\rho : G \to \mathrm{GL}(Z)$ and a latent space $Z = Z_1 \oplus \ldots \oplus Z_K$. Moreover, we assume there exists an embedded submanifold $Z_G \subseteq Z$ such that the action of $G$ on $Z$ restricted to $Z_G$ is *regular*, and $Z_G$ is invariant under the action. Only $Z_G$ will then be used as the codomain for the encoding map, $h : X \to Z_G$.

We demonstrate the assumptions above for the common group structure $G = \mathrm{SO}(2) \times \mathrm{SO}(2)$. For the group representation $\rho = \rho_1 \oplus \rho_2$, with $Z = \mathbb{R}^2 \oplus \mathbb{R}^2$, we can use rotation matrices in $\mathbb{R}^2$ for $\rho_1$ and $\rho_2$. We can then use 1-spheres $S^1 = \{z \in \mathbb{R}^2 : \|z\| = 1\}$ for the embedded submanifold: $Z_G = S^1 \times S^1$. In this case, the action of $G$ on $Z$ restricted to $Z_G$ is indeed *regular*, and $Z_G$ is invariant under the action.

**Unsupervised Learning on Latent Manifold**  To learn encodings only on the latent manifold $Z_G$, we use a Diffusion Variational Autoencoder ($\Delta$VAE) (Perez Rey et al., 2020). $\Delta$VAEs can use any closed Riemannian manifold embedded in a Euclidean space as a latent space (or latent manifold), provided that a certain *projection function* from the Euclidean embedding space into the latent manifold is known and the *scalar curvature* of the manifold is available. The $\Delta$VAE uses a parametric family of posterior approximates obtained from a diffusion process over the latent manifold. To estimate the intractable terms of the negative ELBO, the reparameterization trick is implemented via a random walk.

In the case of $S^1$ as a latent (sub)manifold, we consider $\mathbb{R}^2$ as the Euclidean embedding space, and the projection function[4] $\Pi : \mathbb{R}^2 \to S^1$ normalizes points in the embedding space: $\Pi(z) = z/|z|$. The scalar curvature of $S^1$ is 0.

---

[4]This projection function is not defined for $z = \mathbf{0}$, but this value does not occur in practice.

**Semi-Supervised Learning with Transformation Labels**  Caselles-Dupré et al. (2019) proved that LSBD representations cannot be inferred from a training set of unlabeled observations, but that access to the transformations between data points is needed. They therefore use a training set of observation pairs with a given transformation between them.

However, we posit that only a limited amount of supervision is sufficient. Since obtaining supervision on transformations is typically more expensive than obtaining unsupervised observations, it is desirable to limit the amount of supervision needed.

Therefore, we augment the unsupervised $\Delta$VAE with a supervised method that makes use of transformation-labeled batches, i.e. batches $\{x_m\}_{m=1}^M$ such that $x_m = g_m \cdot x_1$ for $m = 2, \ldots, M$, where the transformations $g_m$ (and thus their group representations $\rho(g_m)$) are known and are referred to as *transformation labels*. The simplified version of the metric from Equation (4) can then be used for each batch as an additional loss term (with $x_0 = x_1$), as it is differentiable under the assumptions described above (using the Euclidean norm).

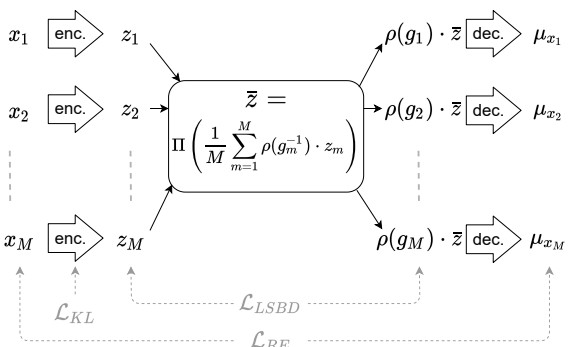

Figure 3: Overview of the supervised part of LSBD-VAE.

We make a small adjustment to Equation (4) for the purpose of our method, since the mean computed there does not typically lie on the latent manifold $Z_G$. Thus, we use the projection $\Pi$ from the $\Delta$VAE to project the mean onto $Z_G$. Writing the encodings as $z_m := h(x_m)$, the additional loss term for a transformation-labeled batch $\{x_m\}_{m=1}^M$ then becomes

$$\mathcal{L}_{LSBD} = \frac{1}{M} \sum_{m=1}^M \left\| \rho(g_m^{-1}) \cdot z_m - \Pi \left( \frac{1}{M} \sum_{m=1}^M \rho(g_m^{-1}) \cdot z_m \right) \right\|^2, \tag{11}$$

where $g_1 = e$, the group identity.

Moreover, instead of feeding the encodings $z_m$ to the decoder, we use $\rho(g_m) \cdot \bar{z}$, where $\bar{z} = \Pi \left( \frac{1}{M} \sum_{m=1}^M \rho(g_m^{-1}) \cdot z_m \right)$. This encourages the decoder to follow the required group structure. This only affects the reconstruction loss component of the $\Delta$VAE.

Figure 3 illustrates the supervised part of our method for a transformation-labeled batch $\{x_m\}_{m=1}^M$. The loss function is the regular ELBO (but with adjusted decoder input as described above) as used in $\Delta$VAE plus an additional term $\gamma \cdot \mathcal{L}_{LSBD}$, where $\gamma$ is a weight hyperparameter to control the influence of the supervised loss component. By alternating unsupervised and supervised training (using the same encoder and decoder), we have a method that makes use of both unlabeled and transformation-labeled observations.

## 6  Experimental Setup

We evaluate the disentanglement of several models on three different image datasets (Square, Arrow, and Airplane) with a known group decomposition $G = \mathrm{SO}(2) \times \mathrm{SO}(2)$ describing the underlying transformations. For each subgroup a fixed number of $|\mathcal{G}_k| = 64$ with $k \in \{1, 2\}$ transformations is selected. The datasets exemplify different group actions of $\mathrm{SO}(2)$: periodic translations, in-plane rotations, out-of-plane rotations, and periodic hue-shifts.

In real settings, not all variability in the data can be modelled by the actions of a group. Therefore, we also evaluate the same models on two datasets ModelNet40 (Wu et al., 2014) and COIL-100 (Nene et al., 1996) that consist of images from various objects (i.e. non-symmetric variation) under known out-of-plane rotations ($\mathrm{SO}(2)$ symmetries). In many settings it is easy to obtain labels for such rotations, e.g. when the camera or object angle is controlled by an agent. See Figure 4 for examples of the datasets. For more details, see the Supplementary Material.

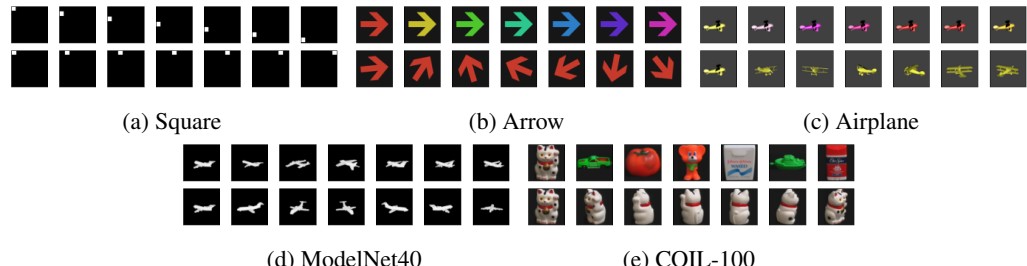

(a) Square            (b) Arrow            (c) Airplane

(d) ModelNet40            (e) COIL-100

Figure 4: Example images from each of the datasets used. Each row shows different examples from a single factor changing.

For the Square, Arrow, and Airplane datasets we test LSBD-VAE with transformation-labeled batches of size $M = 2$. More specifically, for each experiment we randomly select $L$ disjoint pairs of data points, and label the transformation between the data points in each pair. We vary the number of labeled pairs $L$ from 0 (corresponding to a $\Delta$VAE) to $N/2$ (in which case each data point is involved in exactly one labeled pair). We set the weight $\gamma$ of the supervised loss component to $\gamma = 100$ for all experiments. We choose $M = 2$ for our experiments since it is the most limited setting for LSBD-VAE. Higher values of $M$ would provide stronger supervision, so successful results with $M = 2$ imply that good results can also be achieved for higher values of $M$ (but not necessarily vice versa).

For the COIL-100 and ModelNet40 datasets, we train LSBD-VAE on batches containing images of one particular object from all different angles (72 and 64 for COIL-100 and ModelNet40, respectively). Each batch is labelled with transformations $(g_1, e), \ldots, (g_M, e)$, where $g_m$ represent rotations, and the unit transformation $e$ indicates that the object is unchanged. To represent the rotations we use a $S^1$ latent space as in $\Delta$VAE, whereas for the object identity we use a 5-dimensional Euclidean space with standard Gaussian prior as in regular VAEs. LSBD is measured as the disentanglement of rotations in the latent space. For these experiments we used $\gamma = 1$.

We furthermore test a number of known disentanglement methods for comparison, including traditional disentanglement methods as well as methods focusing on LSBD. In particular, we use `disentanglement_lib` (Locatello et al., 2019) to train a regular VAE (Kingma and Welling, 2014; Rezende et al., 2014), $\beta$-VAE (Higgins et al., 2017), CC-VAE (Burgess et al., 2018), FactorVAE (Kim and Mnih, 2018), and DIP-VAE-I/II (Kumar et al., 2018). Furthermore we evaluate the method from Quessard et al. (2020) that focuses on LSBD. We also tested ForwardVAE (Caselles-Dupré et al., 2019), but show only limited results since we were not able to reproduce any reasonable results for our datasets.

We use encodings from all these methods to evaluate $\mathcal{D}_{\text{LSBD}}$, as well as common traditional disentanglement metrics from `disentanglement_lib`: Beta (Higgins et al., 2017), Factor (Kim and Mnih, 2018), SAP (Kumar et al., 2018), DCI Disentanglement (Eastwood and Williams, 2018), Mutual Information Gap (MIG) (Chen et al., 2018), and Modularity (MOD) (Ridgeway and Mozer, 2018).

More information about the architecture, epochs and hyperparameters can be found in the Supplementary Material. For the traditional disentanglement methods trained on Square, Arrow and Airplane datasets the latent spaces have 4 dimensions, since these are the minimum number of dimensions necessary to learn LSBD representations for an underlying $SO(2) \times SO(2)$ symmetry group, see (Higgins et al., 2018; Caselles-Dupré et al., 2019). For COIL-100 and ModelNet40 we use latent spaces with 7 dimensions for a fair comparison with the LSBD-VAE method.

## 7   Results: Evaluating LSBD with $\mathcal{D}_{\text{LSBD}}$

We now highlight four key observations from our experimental results. In particular, we differentiate between the methods (VAE, $\beta$-VAE, CC-VAE, FACTOR, DIP-I, DIP-II) and metrics (BETA, FACTOR, SAP, DCI, MIG, MOD) that approach disentanglement in the *traditional* sense, and methods ($\Delta$VAE, QUESSARD, LSBD-VAE) and metric ($\mathcal{D}_{\text{LSBD}}$) that focus specifically on LSBD. The full quantitative results can be found in the Supplementary Material.

### 7.1 Standard Disentanglement Methods Don't Learn LSBD Representations

Figure 5 summarizes the $\mathcal{D}_{\text{LSBD}}$ scores (lower is better) for all methods on all datasets. Bars show the mean scores over 10 runs for each method, the vertical lines represent standard deviations. LSBD-VAE/$L$ indicates our method trained on $L$ labelled pairs (LSBD-VAE/0 corresponds to the unsupervised $\Delta$VAE), LSBD-VAE/full indicates our method trained on batches containing a single object in all known transformations (for datasets with non-symmetric variation). Note that LSBD-VAE obtained very good scores (nearly 0) on the Arrow and Pixel datasets, hence the missing bars.

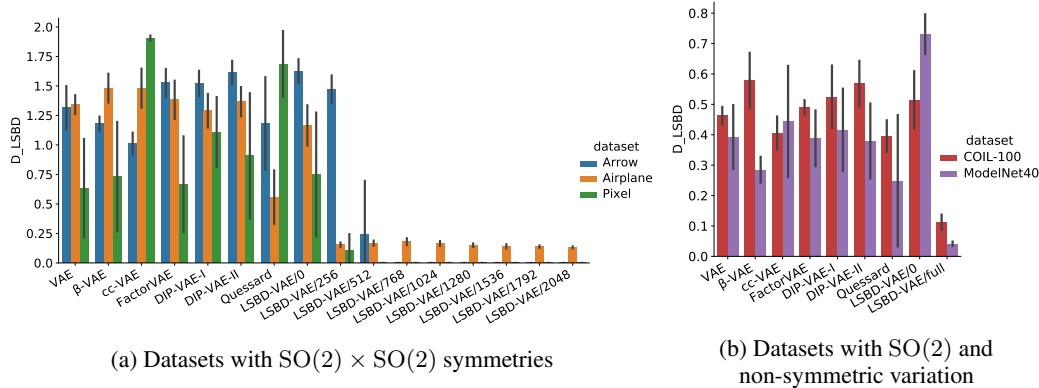

(a) Datasets with $\text{SO}(2) \times \text{SO}(2)$ symmetries

(b) Datasets with $\text{SO}(2)$ and non-symmetric variation

Figure 5: $\mathcal{D}_{\text{LSBD}}$ scores for all methods on all datasets

None of the traditional disentanglement methods achieve good $\mathcal{D}_{\text{LSBD}}$ scores, even if they score well on other traditional disentanglement metrics. This implies that LSBD isn't achieved by traditional methods. Moreover, from the full results in the Supplementary Material we see that the traditional methods on these datasets do not achieve good scores on all traditional metrics. In particular, SAP, DCI, and MIG scores are low. We believe this is a result of the cyclic nature of the symmetries underlying our datasets, further emphasizing the need for disentanglement methods that can capture such symmetries.

The SAP and MIG scores measure to what extent generative factors are disentangled into a single latent dimension. However, since the factors in our dataset are inherently cyclic due to their symmetry structure, they cannot be properly represented in a single latent dimension, as shown by Perez Rey et al. (2020). Instead, at least two dimensions are needed to continuously represent each cyclic factor in our data. A similar conclusion was made by Caselles-Dupré et al. (2019) and Painter et al. (2020).

DCI disentanglement measures whether a latent dimension captures at most one generative factor. This is accomplished by measuring the importance of each latent dimension in predicting the true generative factor using boosted trees. However, since the generative factors are cyclic, the performance of the boosted tree classifiers is far from optimal, thus providing more importance to several dimensions in predicting the generative factors and giving overall lower DCI scores.

### 7.2 LSBD-VAE and other LSBD Methods *Can* Learn LSBD Representations with Limited Supervision on Transformations

From Figure 5 we observe that methods focusing specifically on LSBD can score higher on $\mathcal{D}_{\text{LSBD}}$, showing that they are indeed more suitable to learn LSBD representations. In particular, LSBD-VAE got very good $\mathcal{D}_{\text{LSBD}}$ scores for all datasets. Moreover, our experiments on the Arrow, Airplane, and Pixel datasets also show that only limited supervision suffices to obtain good $\mathcal{D}_{\text{LSBD}}$ scores with low variability.

We only partially managed to reproduce the results from Quessard et al. (2020) on our datasets. Their method scored fairly well on the Airplane, ModelNet40, and COIL-100 datasets, but did not do well on the Square and Arrow dataset in our experiments.

Furthermore, we tested ForwardVAE by Caselles-Dupré et al. (2019), but we did not manage to produce any reasonable results on our datasets, trying both their original architecture and the

architecture we used for our other experiments. Therefore, we do not include scores for this method. We did however manage to reproduce ForwardVAE's results on the Flatland dataset, which was used in their paper. For those experiments, we computed a mean $\mathcal{D}_{\text{LSBD}}$ score of 0.012 with standard deviation 0.001 over 10 runs, indicating that ForwardVAE indeed learns LSBD representations for Flatland.

### 7.3 LSBD Representations Also Satisfy Previous Disentanglement Notions

Our results also indicate that LSBD captures various desirable properties that are expressed by traditional disentanglement metrics. In Figure 6 we compare $\mathcal{D}_{\text{LSBD}}$ scores with scores for previous disentanglement metrics, for all our experiments. Note that for $\mathcal{D}_{\text{LSBD}}$ lower is better, whereas for all other metrics higher is better. As we noted before, good scores on traditional disentanglement metrics don't necessarily imply good $\mathcal{D}_{\text{LSBD}}$ scores. Conversely however, methods that score well on $\mathcal{D}_{\text{LSBD}}$ also score well on many traditional disentanglement metrics, often even outperforming the traditional methods. In particular, from the full results (see Supplementary Material) we see that LSBD-VAE matches or outperforms the traditional methods on the BETA, FACTOR and MOD metrics, and achieves much better scores for the DCI metric where traditional methods scored poorly.

The MIG and SAP scores are still low for methods focusing on LSBD. This is expected however, as explained earlier in Section 7.1. This was also observed by Painter et al. (2020) for different datasets.

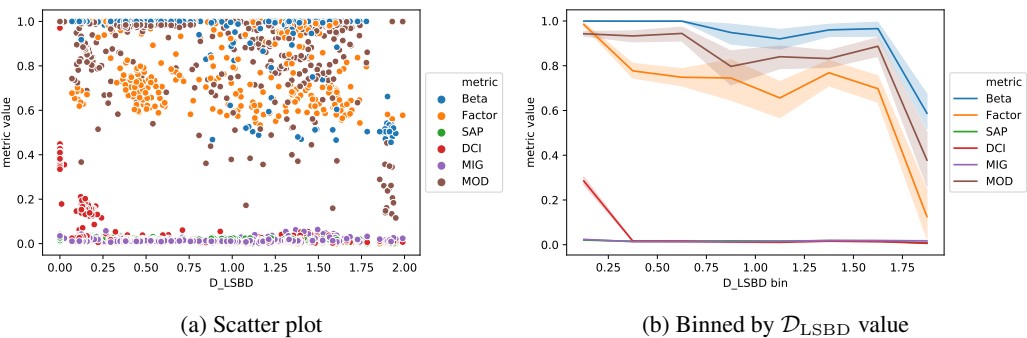

(a) Scatter plot  (b) Binned by $\mathcal{D}_{\text{LSBD}}$ value

Figure 6: Comparing $\mathcal{D}_{\text{LSBD}}$ to previous disentanglement metrics

## 8 Conclusion

We presented $\mathcal{D}_{\text{LSBD}}$, a metric to quantify Linear Symmetry-Based Disentanglement (LSBD) as defined by Higgins et al. (2018). We further used this metric formulation to motivate LSBD-VAE, a semi-supervised method to learn LSBD representations given some expert knowledge on the underlying group symmetries that are to be disentangled.

We used $\mathcal{D}_{\text{LSBD}}$ to evaluate various disentanglement methods, both traditional methods and recent methods that specifically focus on LSBD, and showed that LSBD-VAE can learn LSBD representations where traditional methods fail to do so. We also compared $\mathcal{D}_{\text{LSBD}}$ to traditional disentanglement metrics, showing that LSBD captures many of the same desirable properties that are expressed by existing disentanglement methods. Conversely, we also showed that traditional disentanglement methods and metrics do not usually achieve or measure LSBD.

Challenges that remain are expanding and testing LSBD-VAE and $\mathcal{D}_{\text{LSBD}}$ on different group structures, towards more practical applications, as well as focusing on the utility of LSBD representations for downstream tasks.

**Broader Impact**  The work is fairly theoretical, and practical methods derived from this work have no obvious negative societal impact. However, the ideas presented are relevant to representation learning and could be, in particular, used in computer vision and agent control applications.

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
