# OpenReview forum: "Quantifying and Learning Linear Symmetry-Based Disentanglement"
_NeurIPS.cc/2021/Conference — NeurIPS 2021 Submitted_

### Official Review · Reviewer_CUVD · 2021-07-10

**Rating:** 6
**Confidence:** 4

**Summary:**

This paper proposes a new metric $D_{LSBD}$ to measure Linear Symmetry Based Disentanglement. The metric is motivated from an idealized setting before outlining a practical upper bound that can be realized. The authors also apply their new metric as an additional loss term to a $\Delta$-VAE model and show that it greatly enhances Linear Disentanglement while also agreeing with past notions of Disentanglement. The experiments, while simple and using only $SO(2)$ as a subgroup, are demonstrative of the proposed approach.

**Main Review:**

I really enjoyed the clarity of the presentation in this paper. In particular, using dispersion as a means to define a disentanglement measure is very intuitive and also novel. Moreover, most of the expository figures are well placed and serve as a nice visual guide to much of the theory presented. Overall, I find this paper interesting and a small step towards quantifying linear symmetry based disentanglement.

There are a few comments and mild concerns that I have however which I hope the authors can clarify in their responses. I believe the theory is sound but the assumptions of the theory must be scrutinized. The assumption that the action of $G$ on $X$ is regular is quite a strong assumption and many datasets will have no hope of satisfying this. It would be good to get a quantification of the degradation of performance with respect to the new metric if modify one of the datasets in such a way that the action is no longer free. The second, and perhaps, the biggest gripe I have with this paper is the over reliance on $SO(2)$ as a subgroup. I understand why this subgroup was chosen as it is easy to perform experiments on but much of the practical theory currently does not generalize beyond this group.  However, the paper is currently framed as a more general framework and at the moment it is non-trivial to see how this would be generalized to even simple groups like $E(2)$. As a result, I would ask the author's to either tone down the main claims or repurpose them to be for $SO(2)$.  This to me is the biggest limitation of this work, even though the new disentanglement metric itself is interesting. Moreover, in section 4 the optimization was done over the set of all possible disentangled subspaces ---i.e. finding $\rho$ but in section 5  line 163 for LSBD-VAE one key assumption is that you have a linearly disentangled group representation. This seems at first glance a major decoupling of the theory from practice. It would be great to get some insight into this.

Questions/Minor Remarks:
- Why do we need a change of basis to expose the invariant subspace? Specifically, why can't we consider the total map $h' = (PCA \circ h) (x)$ to be the equivariant encoding function?
- What is the $\rho$ in $X$ space? I think you only mention $\rho$ in $Z$ space
- I think projecting the mean to the latent submanifold is a bit ad hoc. What you really want is to use the Reynolds operator to average over the subgroup.
- Line 58: Not all groups can be decomposed (outside of trivial decompositions). Please amend this statement.
- Line 181: is the scalar curvature is 0 is this not just a flat space rather than S^1? I believe I have a minor misunderstanding here.
- Figure 5, it seems like adding more supervision for the Airplace class (yellowish orange bar) doesn't seem to help as much as for Arrow and Pixel class. Can you maybe shed light into why this is the case?
- Fig 6a is illegible. A table here would be better.
- I would highly recommend an Algorithm box to explain the training procedure of LSBD-VAE. Often it was difficult to decouple the theory from the actual objectives (with the exact equations) being optimized.






**Time Spent Reviewing:**

4 hours

---

> ### Author Response · Authors · 2021-08-10
> **Reply for Reviewer CUVD (1 out of 2)**
>
> We thank the reviewer for the time and effort spent on understanding and reviewing our paper, and for the helpful comments provided.
>
> We are happy that the reviewer enjoyed the clarity of the paper and appreciates its relevance as an incremental step towards the quantification of Linear Symmetry-Based Disentanglement (LSBD). We appreciate the constructive comments provided and we have identified three main topics on which we would like to comment:
>
> - *On the assumptions of the theory: "The assumption that the action of G on X is regular is quite a strong assumption and many datasets will have no hope of satisfying this. It would be good to get a quantification of the degradation of performance with respect to the new metric if modify one of the datasets in such a way that the action is no longer free."*
> 	- We acknowledge that the assumption of a regular action is strong and can restrict the number of datasets where it is applicable. To show a situation where we can relax this assumption, we have presented the experiments with Modelnet40 and COIL-100 datasets, where for each object represented in the dataset there is an orbit corresponding to the object's rotation. Notice that this dataset as a whole does not have a regular group action, however, it is composed of multiple subsets that do. The metric we propose can be estimated for each orbit and can be averaged across all objects. Moreover, the LSBD-VAE method is also capable of handling such situation provided that it is fed with batched data where each batch contains data from the same orbit. This experiment shows that datasets that have countable subsets with regular actions can be both evaluated with our D_LSBD metric and used to create representations with our LSBD-VAE method.  We recognize that we should put extra emphasis on this situation where our conditions can be relaxed.
>
> 	- With respect to datasets where the action is not free. In particular, Higgins et al. [c] in Section 5.2 have discussed the problem of having such datasets. E.g. when having datasets of images from objects with occlusions. In such situations, the problem is that there is no guarantee that suitable action can be defined on the representation space Z such that the encoding function h is equivariant with respect to the group action. A practical solution proposed in Higgins et al. is the use of active sensing. However, it is still a question how LSBD can be measured in these situations, where such practical solutions can't be implemented. We find the suggestion on measuring the degradation of performance of our proposed metric in datasets with non-free group actions something interesting to explore in future work.
>
> - *On the over-reliance of SO(2) group: The second, and perhaps, the biggest gripe I have with this paper is the over reliance on SO(2) as a subgroup. I understand why this subgroup was chosen as it is easy to perform experiments on but much of the practical theory currently does not generalize beyond this group. However, the paper is currently framed as a more general framework and at the moment it is non-trivial to see how this would be generalized to even simple groups like E(2). As a result, I would ask the author's to either tone down the main claims or repurpose them to be for SO(2). "*
>
> 	- We recognize that the practical implementation presented in our work has been restricted to showing examples with SO(2) actions and this should be also clearly reflected in the text, which we will change accordingly to tone down our main claims. However, it is important to note that the proposed metric is general as it is not designed for any particular subgroup. However, the practical computation might indeed require different implementations for different subgroups that are not trivial to generalize and we consider the extension to other subgroups as interesting future work.
>
> - *On the decoupling of theory from practice: "Moreover, in section 4 the optimization was done over the set of all possible disentangled subspaces ---i.e. finding ρ but in section 5 line 163 for LSBD-VAE one key assumption is that you have a linearly disentangled group representation. This seems at first glance a major decoupling of the theory from practice. It would be great to get some insight into this."*
>
> 	- We would like to point out that the objectives of each section are different. In Section 4 we present our general metric for quantifying LSBD. As it has been discussed in the previous point our metric can be quite general and it can be implemented for any subgroup given that suitable parametrizations are used.
> 	- On the other hand, in Section 5 we present the LSBD-VAE method where we assume that some knowledge about the subgroup is known, and we use this knowledge as an inductive bias. This is just a particular method to create LSBD representations, which is inspired by the metric but is not completely derived by it. We would like to emphasize the differences between the metric calculation (and its practical implementation) with respect to the LSBD-VAE method. Each serves a different purpose and we do not necessarily see them as decoupling theory from practice.
>
> To be continued ...

---

> > ### Author Response · Authors · 2021-08-10
> > **Reply for Reviewer CUVD (2 out of 2)**
> >
> > Questions/Minor remarks:
> >
> > - *"Why do we need a change of basis to expose the invariant subspace? Specifically, why can't we consider the total map h′=(PCA∘h)(x) to be the equivariant encoding function?"*
> >
> > 	- With respect to the change of basis, the infimum from Equation 7 requires finding the optimal group representation that is linearly disentangled according to the embedding function. However, for an arbitrary encoding function h such a group representation might not be aligned to the canonical axes, which makes it difficult to identify the subgroup vector subspaces. By finding a suitable change of basis it is then possible to simplify the search for an optimal linearly disentangled group representation since in the new basis the group representation can be expressed as block diagonal matrices where each block corresponds to the group representation of each subgroup.
> > 	- It is true that when calculating the metric, we are implicitly saying the total map h' = PCA∘h is considered to be the equivariant encoding function. We realize that explicitly expressing it this way might make it easier to understand the metric computation.
> >
> > - *"What is the ρ in X space? I think you only mention ρ in Z space"*
> >
> > 	- According to Higgins' definition, an encoding function h is LSBD if there is a disentangled group representation \rho that acts on Z. The definition encourages the linearity of the transformations in the representation space Z. However, in the data space X the group action is not necessarily (and in most cases not) linear, thus we cannot speak about a group representation $\rho$ in X. The group action on X is typically written as a central dot $\cdot$ or as $G_X$. We’ve included a more extensive explanation on the group-theoretic concepts and notations in the supplementary material.
> >
> > - *"I think projecting the mean to the latent submanifold is a bit ad hoc. What you really want is to use the Reynolds operator to average over the subgroup."*
> >
> > 	- Thank you for the suggestion about the Reynolds operator. This was a new concept for us that we did some research on. According to [d] the Reynolds operator R is a group-invariant projection. This would mean that the Reynolds operator R(z) of a vector $z\in Z$ would be invariant to any element of G, i.e. g∘R(z) = R(z). However, the notion of average we are looking for is not invariant to the group actions, e.g. in our case we expect the average to be a point on the circle which is not invariant to the SO(2) group action. However, we recognize we may be missing something important in our understanding so we are open to a discussion on this point.
> >
> > - *"Line 58: Not all groups can be decomposed (outside of trivial decompositions). Please amend this statement."*
> > 	- We realize that our statement was not clearly presented. We meant that the definition of LSBD is given for a group that can be decomposed into K subgroups, but not that this is possible for any group. We will clarify the text to express this, thank you for noticing.
> > - *"Line 181: is the scalar curvature is 0 is this not just a flat space rather than S^1? I believe I have a minor misunderstanding here."*
> >
> > 	- The scalar curvature of an n-sphere with radius r is given by $n(n-1)/r^2$ as described in [e] page 600. This is not to be confused with the sectional curvature given by $1/r^2$, which is constant across different n. Therefore, for S^1 the scalar curvature is 0.
> > - *"Figure 5, it seems like adding more supervision for the Airplace class (yellowish orange bar) doesn't seem to help as much as for Arrow and Pixel class. Can you maybe shed light into why this is the case? "*
> > 	- We would like to bring to notice that our results show that adding supervision does help with the D_LSBD scores for the Airplane dataset, but overall the best achievable D_LSBD value is simply a bit higher. We think the main reason for this is the complexity of the dataset, notice that this also happens with the COIL-100 dataset using the LSBD-VAE/full. This could potentially be improved by choosing a more suitable architecture for the neural networks. However, we have chosen to fix the architecture for all datasets for ease of comparison.
> > - *"Fig 6a is illegible. A table here would be better"*
> > 	- We have included a table in the supplementary material containing all the relevant information, but overall we don’t find it very easy to read. Thus, we have included aggregating figures such as Fig6a and b to summarize those results. However, we recognize that we can improve the legibility of Fig6a and we will provide a better explanation of its meaning. We do believe that it helps in understanding the more aggregated Fig 6b, which is one of the reasons we included it.
> > - *"I would highly recommend an Algorithm box to explain the training procedure of LSBD-VAE. Often it was difficult to decouple the theory from the actual objectives (with the exact equations) being optimized."*
> > 	- We will include such an Algorithm box in the supplementary materials to help clarify the explanations provided.
> >
> > ### References
> > [c] Higgins, I., Amos, D., Pfau, D., Racaniere, S., Matthey, L., Rezende, D., and Lerchner, A. (2018).Towards a definition of disentangled representations, arXiv preprint, arXiv:1812.02230
> >
> > [d] Derksen, H., and Kemper, G., (2015) "Computational Invariant Theory", Chapter 2: Invariant theory, Springer.
> >
> > [e] Joachim, M., and Wraith, D. (2008). "Exotic spheres and curvature". In Bulletin of the American Mathematical Society, 45(4), 595-616.

---

> > > ### Comment · Reviewer_CUVD · 2021-08-19
> > > **Thank you for your response**
> > >
> > > Thank you for your thoughtful response to my questions. I believe most of my questions have been suitably answered and I am more confident in maintaining my score of a weak accept.

---

### Official Review · Reviewer_KFfN · 2021-07-15

**Rating:** 5
**Confidence:** 3

**Summary:**

This work focuses on developing a metric evaluation metric and a semi-supervised VAE-based model for linear symmetry-based disentanglement (LSBD)[1]. When assuming knowledge of linearly disentangled group representation $\rho$, the authors deduce that LSBD is fulfilled if the last requirement is met (equivariance of a map h). The authors propose to measure the equivariance by simply quantify the equivariance of the encoding map w.r.t. to the group and observations. Although a straightforward thought, computing the metric is challenging. Therefore, the authors propose an approximation by calculating the dispersion of the inverse group representation obtained through a PCA. Similar to the evaluation metric, the authors propose LSBD-VAE, a semi-supervised version of $\Delta$-VAE to regularise the latent space based on its dispersion. They show through empirical evaluation that their method outperforms other unsupervised disentanglement approaches w.r.t. to the proposed evaluation metric.


**Limitations And Societal Impact:**

The work does not discuss any potential societal impact. Adding such a discussion would improve the paper's overall quality. For instance, in that discussion, the authors could discuss both positive (interpretability, implications for understanding and evaluating fairness/bias in models) and negative impact (using representations for discriminating against specific subgroups and thus, compromise fairness) of disentangled representation learning.

**Main Review:**

_Originality, significance, and general assessment_

The symmetry-based approach to disentanglement is theoretically sound and appealing. However, the work did not provide a proposal on learning the appropriate representation that fulfills the group-based disentanglement requirements. This work tries to fill this gap by proposing the LSBD evaluation metric and semi-supervised VAE. Although I found this research direction very exciting, I have some concerns about this work:
* Strong assumptions which may not hold beyond synthetic data: This work assumes explicit knowledge of the data and its structure for the LSBD evaluation metric and method. For each data pair, we need to know the corresponding group and linear transformation. This is quite a strong assumption, even if the semi-supervised version only assumes knowledge of some thousands of data pairs. I worry that we cannot judge from this work whether this is a realistic setting and whether some thousands of data pairs will be enough to generalize in a non-synthetic setting.
* Fair benchmark comparison: LSBD-VAE has explicit knowledge of the groups of pairs of data instances. Even though only optimized in a semi-supervised fashion, this gives the optimization more signals than in an unsupervised setting. Therefore, comparing LSBD-VAE to only unsupervised methods is not fair. Locatello et al.[2] proposed a weakly-supervised approach that might be a more suitable comparison.
* Applicability: I do think with disentanglement, it is not about just perfectly learning to disentangle the factors of variation, but also about the promise that disentangled representations generalize better, are fairer, and more robust. This work only shows that the model can disentangle well w.r.t. the evaluation metric they proposed. However, the part of generalization, e.g., reconstruction ability and performance on downstream tasks, has not been investigated.
* Usefulness: The big question that remains unanswered after reading this paper is: "Why should I use this metric?" It is not clear why this evaluation metric is superior to the other 5-6 existing ones. What do other metrics fail to capture, which the LSBD can? Can you show some examples, theoretically or empirically, where the other metrics but LSBD fail?

_Quality and clarity_

In general, I could follow the paper and its structure. W.r.t. related work and the base model that LSBD is using, I think the authors should elaborate more details to make the paper self-contained:
* Related work (l.47-51): This discussion is very vague and does not describe what these works do. Can you elaborate on these works and how they are different from the semi-supervised approach?
* $\Delta$-VAE was used as the base for the semi-supervised approach. However, it was barely described. For myself, it helped reading through the original paper to have a better understanding of what precisely the method does. However, for the work to be self-contained, it might be helpful to the reader to have a brief description of $\Delta$-VAE.

_Comments and questions_

* Figure 6 & ll.316-319: I do not necessarily agree with "good scores on traditional disentanglement metrics do not necessarily imply good $D_{LSBD}$ scores". In particular, Figure 6b) shows that Beta, Factor, and MOD scores are proportional to $D_{LSBD}$. High Beta, Factor, and MOD scores correspond to lower $D_{LSBD}$, and lower Beta, Factor, and MOD scores correspond to higher. Can you elaborate?
* LSBD evaluation metric: Except for the difference in the "exact" values, it is not clear at all why your evaluation metric should be preferred to the other ones?
* What is the percentage of labels used (e.g., for the results plotted in Figure 5)? You only mentioned the absolute number of labels, not the relative fraction.
* All baseline disentanglement models are trained in an unsupervised manner. In contrast, the proposed model knows the groups and uses that in a semi-supervised fashion. I think a fairer comparison would be [3] as it uses weak supervision between pairs of data. This is quite similar to the way LSBD-VAE uses data pairs with the knowledge of its group and transformation. In my opinion, this work can be seen as a weakly-supervised approach. The paper might improve from discussing the similarities with [2].

**Time Spent Reviewing:**

5

---

> ### Author Response · Authors · 2021-08-10
> **Reply for Reviewer KFfN**
>
> We thank the reviewer for the time and effort spent on understanding and reviewing our paper, and for the helpful comments provided.
>
> We are happy that the reviewer is excited about our research direction and acknowledges the theoretical soundness of group-based disentanglement. We will respond to the main points in the review individually.
>
> We have provided a general comment that addresses concerns that were mentioned by multiple reviewers, and will refer to parts of this comment where relevant.
>
> With respect to the concerns:
> - *"Strong assumptions which may not hold beyond synthetic data: This work assumes explicit knowledge of the data and its structure for the LSBD evaluation metric and method. For each data pair, we need to know the corresponding group and linear transformation. This is quite a strong assumption, even if the semi-supervised version only assumes knowledge of some thousands of data pairs. I worry that we cannot judge from this work whether this is a realistic setting and whether some thousands of data pairs will be enough to generalize in a non-synthetic setting."*
>
> 	- It is true that a group decomposition needs to be known, but this is a general property of LSBD as proposed by Higgins et al. [c], not just of our metric/method. For many common groups, possible linear representations are known. While we agree that this limits the scope of our paper, we think our contributions fit the scope of the general LSBD setting, in particular the D_LSBD metric. The LSBD-VAE method requires somewhat stronger assumptions indeed, but mainly serves to show further utility of the metric formulation, and to show one way to learn LSBD representations. We believe our contributions are important to open up possibilities for further research into LSBD methods, thereby filling a gap that was left after the original LSBD paper by Higgins et al. [c].
>
> - *"Fair benchmark comparison: LSBD-VAE has explicit knowledge of the groups of pairs of data instances. Even though only optimized in a semi-supervised fashion, this gives the optimization more signals than in an unsupervised setting. Therefore, comparing LSBD-VAE to only unsupervised methods is not fair. Locatello et al.[2] proposed a weakly-supervised approach that might be a more suitable comparison."*
>
> 	- Please see our general comment on the comparison with unsupervised/semi-supervised methods (2). We have provided a comparison to standard unsupervised disentanglement methods to show the contrast on how our metric and method relate to previous notions of disentanglement. Notice that we have provided results obtained from the method by Quessard et al. [a], which also uses a kind of (semi-)supervision on transformations and fits the paradigm of LSBD.
> 	- Unfortunately, it seems that the list of references was not included in the provided review. We are very interested in evaluating whether the corresponding reference [2] could be used for comparison.
>
> - *"Applicability: I do think with disentanglement, it is not about just perfectly learning to disentangle the factors of variation, but also about the promise that disentangled representations generalize better, are fairer, and more robust. This work only shows that the model can disentangle well w.r.t. the evaluation metric they proposed. However, the part of generalization, e.g., reconstruction ability and performance on downstream tasks, has not been investigated."*
>
> 	- We thank the reviewer for the comment provided, we recognize the importance of showing some applications of LSBD to highlight its importance. However, the focus of this work has been centered on the development and description of a metric to quantify the disentanglement and a method derived from it. In the introduction of our paper, we refer to the work in [b], which has evaluated the use of LSBD on the downstream task of predicting the action between two consecutive observations as proof of its importance. We would like to further explore the applications of LSBD as future work.
>
> - *"Usefulness: The big question that remains unanswered after reading this paper is: "Why should I use this metric? It is not clear why this evaluation metric is superior to the other 5-6 existing ones. What do other metrics fail to capture, which the LSBD can? Can you show some examples, theoretically or empirically, where the other metrics but LSBD fail?"*
> 	- Part of the usefulness of the LSBD paradigm lies in the fact that it exploits the structure induced by group transformations in the real world from which data is collected. Other disentanglement metrics try to separate the different factors, which might not preserve this structure. Our experiments show those cases where other metrics fail. In particular, for factors with a cyclic nature (typically induced by a symmetry group), previous metrics do not capture whether this structure is preserved. Our results show that good scores for previous metrics do not necessarily imply good scores for D_LSBD (see Section 7.1), even though vice versa this typically is the case (as shown in Section 7.3).
> Please see also the first and second point under "Comments and questions" of this reply.
>
> Quality and clarity
>
> - *"Related work (l.47-51): This discussion is very vague and does not describe what these works do. Can you elaborate on these works and how they are different from the semi-supervised approach?"*
> 	- Due to space constraints, we have chosen to keep the related work very concise. We can add a short paragraph that summarizes the distinctive traits that characterize the related work. We will add a more detailed explanation on the supplementary material of the most relevant related work.
> - *"Δ-VAE was used as the base for the semi-supervised approach. However, it was barely described. For myself, it helped reading through the original paper to have a better understanding of what precisely the method does. However, for the work to be self-contained, it might be helpful to the reader to have a brief description of Δ-VAE."*
> 	- We, unfortunately, had to cut a more detailed description of Delta-VAE due to space constraints, but we will include it in the supplementary material instead.
>
> Comments and Questions
>
> - *"Figure 6 & ll.316-319: I do not necessarily agree with "good scores on traditional disentanglement metrics do not necessarily imply good DLSBD scores". In particular, Figure 6b) shows that Beta, Factor, and MOD scores are proportional to DLSBD. High Beta, Factor, and MOD scores correspond to lower DLSBD, and lower Beta, Factor, and MOD scores correspond to higher. Can you elaborate?"*
> 	- Figure 6a and b show many examples with high Beta/Factor/MOD scores that nevertheless get bad D_LSBD scores (e.g. higher than 1). Even though there is some correlation, there are also plenty of examples that do not follow this trend, hence our conclusion that good scores don't necessarily imply good D_LSBD scores. We remark in particular that D_LSBD scores going from 0 towards 1.5 (a significantly low score) show a little decrease in value, especially for Beta and MOD scores.
>
> - *"LSBD evaluation metric: Except for the difference in the "exact" values, it is not clear at all why your evaluation metric should be preferred to the other ones?"*
> 	- This relates to the previous point discussed, as well as our reply to the usefulness of the metric. LSBD captures structure induced in the data by group transformations that traditional disentanglement cannot measure. We build our work on the motivation provided by Higgins et al. [c] for the need of a proper formalised definition of disentanglement that exploits group structure from the real world. We agree that we could emphasise this point more clearly in our own work as well, and we will gladly do so.
>
> - *"What is the percentage of labels used (e.g., for the results plotted in Figure 5)? You only mentioned the absolute number of labels, not the relative fraction."*
> 	- Thank you for this suggestion, we realize that including this information can help understand the dimensions of the supervision used. For a more detailed explanation on the amount of supervision used, please see our general comment on the amount of supervision needed for LSBD-VAE (3), where we provide a discussion on the label percentages.
>
> - *"All baseline disentanglement models are trained in an unsupervised manner. In contrast, the proposed model knows the groups and uses that in a semi-supervised fashion. I think a fairer comparison would be [3] as it uses weak supervision between pairs of data. This is quite similar to the way LSBD-VAE uses data pairs with the knowledge of its group and transformation. In my opinion, this work can be seen as a weakly-supervised approach. The paper might improve from discussing the similarities with [2]."*
> 	- Please see our general comment on the comparison with unsupervised/semi-supervised methods (4). Unfortunately, it seems that the list of references was not included in the provided review, we are very interested in the cited references we could compare to.
>
> ### References
> [a] Quessard, R., Barrett, T. D., and Clements, W. R. (2020). Learning group structure and disentangled representations of dynamical environments. In Advances in Neural Information Processing Systems, 33.
>
> [b] Caselles-Dupré, H., Ortiz, M. G., and Filliat, D. (2019). Symmetry-based disentangled representation learning requires interaction with environments.  In Advances in Neural Information Processing Systems, 4606–4615.
>
> [c] Higgins, I., Amos, D., Pfau, D., Racaniere, S., Matthey, L., Rezende, D., and Lerchner, A. (2018).Towards a definition of disentangled representations, arXiv preprint, arXiv:1812.02230.

---

> > ### Comment · Reviewer_KFfN · 2021-08-10
> > **Missing references**
> >
> > Hi all,
> >
> > apologies for the missing references. I missed to add them when copy&paste the reviews to openreview. Here they are:
> >
> > [1] Higgins, I., Amos, D., Pfau, D., Racaniere, S., Matthey, L., Rezende, D. and Lerchner, A., 2018. Towards a definition of disentangled representations. arXiv preprint arXiv:1812.02230.
> >
> > [2] Locatello, F., Poole, B., Rätsch, G., Schölkopf, B., Bachem, O. and Tschannen, M., 2020, November. Weakly-supervised disentanglement without compromises. In International Conference on Machine Learning (pp. 6348-6359). PMLR.
> >
> > [3] Shu, R., Chen, Y., Kumar, A., Ermon, S. and Poole, B., 2019. Weakly supervised disentanglement with guarantees. ICLR 2020.
> >
> > Thanks for replying to my review. I'll have a look!

---

> > > ### Comment · Reviewer_KFfN · 2021-08-25
> > > **Response to the rebuttal**
> > >
> > > Hi,
> > >
> > > first of all, thank you for the detailed response to my questions. Even though the paper makes a worthy contribution to propose the LSBD metric is, I still have doubts about the proposed model and the evaluation part. I am aware, that the main contribution is a way to evaluate LSBD, however, one of the main usage is to evaluate disentanglement methods. I feel like incorporating more complex datasets and comparing LSBD-VAE to weakly-supervised models would show more impact. Therefore, I am keeping my original score.

---

### Official Review · Reviewer_j9A8 · 2021-07-16

**Rating:** 7
**Confidence:** 4

**Summary:**

Summary:
This paper aims to operationalize the symmetry-based disentanglement idea proposed in Higgins et al., 2018, and proposes a novel loss function that can be used to both evaluate and learn disentangled representations (including learning from datasets where only a subset of samples are fully labeled). The paper includes some experimental validation of the proposed ideas on synthetic datasets. It also includes cross-comparisons between both the LSBD-based metric and method to state of the art disentangling metrics and methods, which demonstrate the complementary aspects of the LSBD method (e.g., its ability to capture multi-dimensional disentangled factors, for example rotations in a euclidean plane).

Overall, the paper is clearly written and makes a positive contribution in pushing for disentangling metrics that are motivated by a novel formalism. The impact of this paper is slightly limited due to the strength of supervision required (full supervision of all factors of variation) for implementing both the D_LSBD metric and the loss.

**Ethical Concerns:**

No concerns

**Limitations And Societal Impact:**

The authors have adequately addressed limitations and potential negative impact

**Main Review:**

Originality
* Novel method for evaluating and learning disentangled representations.
* The objective function can be trained in a semi-supervised method with limited fully-supervised data (although the level of supervision required is extremely high)
* You have done thorough comparison to unsupervised disentanglement approaches, but it would also be useful to know how your method fares against standard semi-supervised approaches, e.g., [1], which could be adapted for a continuous supervised variable.

Quality/Clarity
* There are multiple ways of representing different transformations. For example, rotations can be represented either as 2D coordinates (as you have assumed here) or as 1D angles. In the case of unsupervised disentangling, it’s often unclear what the most parsimonious representation should be - how could the uncertainty over representations be handled in your framework?
* For your LSBD-VAE: It would be interesting to know are the relative contributions of L_{LSBD} and the adjusted likelihood (reconstructing from $\rho(g_m) \cdot \bar{z}$ rather than z_m) to the disentangling performance in the empirical results section.

Significance
* The supervised set must have full labels for all factors/groups. If even one factor is missing, then this method cannot be applied, since we do not know the full set of transformations that relate different data points to each other. It is not clear to me how this assumption can be relaxed with the proposed metric/objective: if we lack labels for only a single factor, then it is no longer possible to explain the dataset in terms of actions on a single input sample.
* The above also means that experiments can only be done on synthetic datasets, where all factors are known.

References
[1] Kingma, Diederik P., et al. "Semi-supervised learning with deep generative models." Advances in neural information processing systems. 2014. Kingma, D. P., Mohamed, S., Rezende, D. J., & Welling, M. (2014). Semi-supervised learning with deep generative models. In Advances in neural information processing systems (pp. 3581-3589)

**Time Spent Reviewing:**

4

---

> ### Author Response · Authors · 2021-08-10
> **Reply for Reviewer j9A8**
>
> We thank the reviewer for the time and effort spent on understanding and reviewing our paper, and for the helpful comments provided.
>
> We are happy that the reviewer acknowledges the importance of disentanglement metrics with more formalism. We will respond to the main points in the review individually.
>
> Originality:
>
> - *"The objective function can be trained in a semi-supervised method with limited fully-supervised data (although the level of supervision required is extremely high)"*
> 	- We refer to our general comment about transformation labels (1) and the amount of supervision for LSBD-VAE (2) for a more detailed response. We agree that a certain level of supervision is crucial for good performance, though we don't fully agree that the level of supervision required is "extremely high".
> - *"You have done thorough comparison to unsupervised disentanglement approaches, but it would also be useful to know how your method fares against standard semi-supervised approaches, e.g., [1], which could be adapted for a continuous supervised variable."*
>
> 	- We refer to our general comment about comparison with other methods (2).
>
> Quality/Clarity:
>
> - *"There are multiple ways of representing different transformations. For example, rotations can be represented either as 2D coordinates (as you have assumed here) or as 1D angles. In the case of unsupervised disentangling, it’s often unclear what the most parsimonious representation should be - how could the uncertainty over representations be handled in your framework?"*
>
> 	- We specifically deal with *Linear* Symmetry-Based Disentanglement, which cannot be achieved when rotations are represented as 1D angles (in latent space). Thus, for LSBD this uncertainty does not exist. This in fact highlights an advantage of the LSBD definition, as it requires models to maintain the cyclic structure of angles (that is lost when 1D angle values are used).
> - *"For your LSBD-VAE: It would be interesting to know are the relative contributions of L_{LSBD} and the adjusted likelihood (reconstructing from $\rho(g_m)\cdot\bar{z}$ rather than z_m) to the disentangling performance in the empirical results section."*
>
> 	- This is indeed a good point, thank you for the suggestion. We did not include a comparison between reconstructing from $z_m$ versus $\rho(g_m)\cdot\bar{z}$. Although we don't think this is very important to the main point of our paper, we'd be happy to include such a comparison in the supplementary material, as it would indeed be interesting for practical purposes.
>
> 	- As for the contribution of L_{LSBD}, the comparison with DeltaVAE (or LSBD-VAE with L=0) exactly shows this.
>
> Significance:
>
> - *"The supervised set must have full labels for all factors/groups. If even one factor is missing, then this method cannot be applied, since we do not know the full set of transformations that relate different data points to each other. It is not clear to me how this assumption can be relaxed with the proposed metric/objective: if we lack labels for only a single factor, then it is no longer possible to explain the dataset in terms of actions on a single input sample."*
> 	- It is indeed true that currently all transformation labels should be known, but it should be easy to relax this requirement. If one factor is missing, we can simply not give any learning signal for those latent subspaces for which we are missing transformation-labels. Similarly, the metric can be computed for a subset of the subgroups as well by projecting onto the corresponding latent spaces (although of course this metric will only quantify those subgroup transformations that are given). We did not investigate this further as it is outside of the scope of this paper.
> - *"The above also means that experiments can only be done on synthetic datasets, where all factors are known"*
>
> 	- Please see our general comment on toy datasets (4). This claim is true for computing the D_LSBD metric properly, but we remark that this is the case for every single disentanglement metric that currently exists. One of the main goals for the metric is benchmarking, in which case it's reasonable to assume these labels are present. The LSBD-VAE method however gives a way to achieve LSBD with fewer labels available. As explained above, even if not all factors are known it is still possible to relax the method to deal with this (though how well this performs would need further experimentation).

---

### Official Review · Reviewer_UCXu · 2021-07-18

**Rating:** 6
**Confidence:** 4

**Summary:**

This paper makes two contributions:
1 - It presents a framework for measuring linear symmetry based disentanglement (LSBD). In particular, it addresses the challenge that LSBD is only defined with respect to an optimal decomposition of the latent space $Z = Z_1 \oplus \ldots \oplus Z_k$ that needs to be identified by the metric.
2 - The metric serves as additional supervision in the training of a VAE on toy datasets with at most two symmetry transformations. The results show quantitative improvements in terms of disentanglement.

**Limitations And Societal Impact:**

The authors have satisfyingly addressed this question.

**Main Review:**

## Recommendation
I am recommending to accept this submission which offers solid mathematical treatment
- the background, problem and goal of the metric are clearly and intuitively explained
- the dispersion approach promises to reduce the level of transformation supervision needed by the metric
- section 4.2 has some very nice aspects, in particular (i) extending to non-uniform sample distributions via a reparametrization of the measure using a transitivity assumption is very rigorous and elegant (ii) the final definition of the metric in equation (7) is clean and general.
- the practical implementation in section 4.3 is sound

The recommendation is weak because I see the following limitations to the significance the approach:
- the LSBD-VAE seems to make the assumption that the latent state decomposition $Z = Z_1 \oplus \ldots \oplus Z_k$ is now a given of the problem, which simplifies differentiability of the metric and seems to contradict the prior definitions. I would argue that this diminishes the originality of the method. Also, this would explain the results described in 7.3 that "LSBD Representations Also Satisfy Previous Disentanglement Notions"
- despite the claims of limited supervision, I believe the amount of supervision used (transformation-labeled batches, l193) is important, which explains why LSBD-VAE outperforms previous approaches so clearly.
- the experimental evaluation is restricted to toy datasets.


## Remarks
- The PCA step (l139-143) is done independently for each subgroup. What happens if different subgroups are projected on the same dimensions. Is this sound?
- Is the upper bound on l155 theoretically proven?

**Time Spent Reviewing:**

4

---

> ### Author Response · Authors · 2021-08-10
> **Reply for Reviewer UCXu**
>
> We thank the reviewer for the time and effort spent on understanding and reviewing our paper, and for the helpful comments provided.
>
> We are happy that the reviewer appreciates the key elements of our paper, such as the mathematical formalisation and generality, the dispersion approach to quantifying and learning LSBD, and the practical implementation. We also appreciate the constructive feedback on the significance of the approach.
>
> We have provided a general comment that addresses concerns that were mentioned by multiple reviewers and will refer to parts of this comment where relevant.
>
> In response to the mentioned limitations:
> - *"the LSBD-VAE seems to make the assumption that the latent state decomposition Z=Z1⊕…⊕Zk is now a given of the problem, which simplifies differentiability of the metric and seems to contradict the prior definitions. I would argue that this diminishes the originality of the method. Also, this would explain the results described in 7.3 that "LSBD Representations Also Satisfy Previous Disentanglement Notions"*
>
> 	- We realise that the phrasing in lines 163-164 is a bit unfortunate, we will update this. The latent space decomposition is not so much a given of the problem (although the group decomposition is, per the definition of LSBD), but setting the latent space decomposition is part of the LSBD-VAE method and can be seen as an inductive bias or expert knowledge. Of course, we do require the problem to allow for a suitable latent space decomposition to exist.
>
> 	- This is indeed a simplification w.r.t. the D_LSBD metric, but our goal for the LSBD-VAE method is to find some specific LSBD representation, thus it does not need to be as general as the metric (nor would this be desirable). Therefore we don't see a contradiction here.
> 	- We agree that the choice of latent space decomposition influences the results from section 7.3, since this leads to more axis-aligned disentangled representations. This is a very good point that we could explain further in the paper. However, we believe a comparison between LSBD and traditional disentanglement is only sensible in this axis-aligned setting. Otherwise, it would make more sense to find invariant subspaces first as in our D_LSBD metric, and compute the traditional disentanglement metrics given the new basis instead.
> - *"despite the claims of limited supervision, I believe the amount of supervision used (transformation-labeled batches, l193) is important, which explains why LSBD-VAE outperforms previous approaches so clearly."*
>
> 	- It is indeed true that the supervision is important for good results. We refer to our general comments on transformation labels (1), comparison with previous approaches (2), and amount of supervision (3) for a detailed discussion. While the supervision is indeed a key element for LSBD-VAE to perform well, we do believe our supervision is fairly limited, and that the difference between the D_LSBD scores is mainly due to the different paradigm between LSBD and previous disentanglement notions, rather than just due to supervision.
>
> - *"the experimental evaluation is restricted to toy datasets."*
> 	- Please see our general comment on toy datasets (4).
>
> In response to the remarks:
>
> - *"The PCA step (l139-143) is done independently for each subgroup. What happens if different subgroups are projected on the same dimensions. Is this sound?"*
>
> 	- This is a good point indeed. We should furthermore require that the eigenvectors from the PCA steps are all linearly independent, otherwise, the results of the metric will not be sound. We will add this requirement to the paper.
>
> 	- However, we believe that in practice it'll essentially never happen that the dimensions for subgroups overlap (i.e. that some eigenvectors are linearly dependent). Firstly, because small numerical differences will suffice for linear independence (although admittedly in close cases this may lead to numerical instability). And secondly, e.g. VAE-based models also include a reconstruction term, that wouldn't allow for multiple subgroups to be represented in the same subspace.
> - *"Is the upper bound on l155 theoretically proven?"*
> 	- Another fair point, we did not explicitly explain why this is an upper bound, we will rectify this. It is an upper bound simply because we are computing the expression for a particular rho, whereas the true metric is the infimum over all possible rho's, which is by definition lower or equal.

---

### Author Response · Authors · 2021-08-10
**General Response for Similar Comments**

We thank all reviewers for their time and effort spent on reviewing our paper, and for the many useful comments. We are glad to see that they mostly share our excitement for this research direction.

There are a few comments that came up multiple times, or that relate to similar topics. We would like to clarify some points in a general comment, which we'll refer to in our replies to individual reviewers. We will gladly update our text to clarify these points in the paper as well wherever needed.

**1. Transformation labels & amount of supervision.**

First, we would like to emphasise that although we speak about supervision, our notion of "transformation labels" is quite different from the more commonly used "factor labels" in disentanglement (or labels in traditional [semi-]supervised learning settings). In particular, our tranformation labels contain information about relative differences between images (pairs of images in our examples), but no absolute information about any particular image. This has some consequences for the interpretation of our semi-supervised results, in particular we believe the supervision is weaker than it may appear.

Technically, in a dataset of N examples, a total of N choose 2 transformation labels could be given between pairs of data points, although under our assumptions we show that only N suffice to fully describe the data w.r.t. transformations (see footnote 2, page 3). Thus, even if all N data points are involved in exactly one labelled pair, we use only N/2 labels. In particular this means that all pairs are still disjoint, i.e. we don't know the transformations between any of these pairs. So even our LSBD-VAE/2048 and LSBD-VAE/full settings in Figure 5 are not actually fully labelled, they only use half of all possible labels.

Moreover, since transformation labels only hold relative information about the relationship between data points, and we only label disjoint pairs of data points, this paradigm provides weaker information than the typical "factor labels" paradigm, where each label puts a data point in perspective with all other (labelled) data points.

**2. Comparison with unsupervised/semi-supervised methods**

Multiple reviewers remarked that we only compare our LSBD-VAE to unsupervised disentanglement metrics. We understand this concern, but would like to clarify that this isn't entirely true, and explain why we compare with traditional unsupervised disentanglement methods.

First, we emphasise that we compare to Quessard et al.'s [a] method and ForwardVAE [b] as well (although we couldn't reproduce good results on our datasets for ForwardVAE), which do use a kind of (semi-)supervision on transformations that fits in our LSBD/transformation-labels paradigm. In particular, they consider only a limited number (4 in most examples) of "atomical" transformations that can be observed between two consecutive data points, and they assume that these are always known.

Second, we refer to the explanation above about transformation labels vs. factor labels, most semi-supervised disentanglement methods operate on this different paradigm of factor labels, thus we do not think they make for a sensible comparison with our semi-supervised method (whereas Quessard et al. [a] and ForwardVAE [b] do).

Third, we want to clarify that the main purpose of our LSBD-VAE method is to show another utility of the formulation in our D_LSBD metric and to show that LSBD representations can be learned reliably this way (as measured by our D_LSBD metric). The most important contribution in our paper however is the D_LSBD metric that allows to even measure LSBD performance at all, and to use it to show how LSBD related to previous notions of disentanglement. This is the main purpose of the unsupervised disentanglement methods in our results.


**3. Amount of supervision needed for LSBD-VAE**

Multiple reviewers comment that our LSBD-VAE method needs a lot of supervision to work, and that the D_LSBD metric requires full knowledge about the transformations. This is mostly true, though we'd like to explain why we think this is not a big shortcoming.
As for the number of labels needed for LSBD-VAE, we remark that for our SO(2)xSO(2) datasets that have N=4096 images (this is explicitly mentioned in the suppl. mat. but not in the main paper, we will rectify this) we generally obtain good results with 256 or at most 512 transformation labels. This means that 12.5% respectively 25% of all images are involved in some labeled pair, but as explained in our first comment this corresponds to only 6.25% respectively 12.5% of all possible transformation labels. As further detailed in our first comment, we believe this is actually not a lot of supervision, though we admit that supervision is very important to achieve good results.

Moreover, since transformation labels require only relative information between data points, we believe there are many practical settings in which such labels can be easily or cheaply obtained. In particular, in agent-environment settings, an agent's actions can provide transformation labels for the observations before and after such actions.

Finally, in practical settings representations are often learned on auxiliary datasets and then transferred to the target domain. Our LSBD-VAE method can train versatile representations on synthetic datasets, for which it is easier to obtain transformation labels.


**4. Full supervision needed for D_LSBD, toy datasets**

Our D_LSBD metric does indeed need full label information to be computed, but we remark that this is the case for every disentanglement metric currently in existence. The main purpose of these metrics is benchmarking, which is typically done with datasets that have full label information (often toy datasets). However, our LSBD-VAE results suggest that computing the D_LSBD metric with fewer known labels can already provide a good proxy for the LSBD performance. We believe that further extensive experimentation with real-life datasets is beyond the scope of this paper, though a very interesting future research direction indeed. Lastly, we remark that all previous methods focusing on LSBD have used similar toy datasets, mostly of lower complexity (such as the Flatland dataset in [a] and [b]).

### References
[a] Quessard, R., Barrett, T. D., and Clements, W. R. (2020). Learning Group Structure and Disentangled Representations of Dynamical Environments. In Advances in Neural Information Processing Systems, 33.

[b] Caselles-Dupré, H., Ortiz, M. G., and Filliat, D. (2019). Symmetry-based disentangled representation learning requires interaction with environments. In Advances in Neural Information Processing Systems, pages 4606–4615.

---

### Decision · Program_Chairs · 2021-09-27

**Decision:**

Reject

**Comment:**

The reviewers agreed that the proposed metric for linear symmetry based disentanglement is solid.  However, the three main criticisms were:
1) The lack of comparison with semi-supervised or weakly-supervised alternative methods, considering that this method requires labeled data,
2) The lack of empirical checking that LSBD correlates with downstream properties that are more directly useful in concrete tasks, and
3) The fact that SO(2) was the only group investigated in depth in the paper, despite the presentation as a general method.

Overall, the paper is not in bad shape, but the limited cope of applicability and empirical comparisons mean it is slightly below the bar.  Other examples of symmetry groups, correlation with metrics such as robustness, and comparison with semi-supervised methods would all substantially improve the paper.